# Spatial–Temporal Characteristics of Precipitation and Its Relationship with Land Use/Cover Change on the Qinghai-Tibet Plateau, China

**Bo Zhang** [1] and **Wei Zhou** [1,2,*]

1 School of Land Science and Technology, China University of Geosciences Beijing, Beijing 100083, China; zhang.bo@cugb.edu.cn
2 Land Consolidation and Rehabilitation Center, Ministry of Natural Resources, Beijing 100035, China
* Correspondence: zhouw@cugb.edu.cn; Tel.: +86-010-8232-1867

**Abstract:** The Qinghai-Tibet Plateau (QTP) is an area sensitive to global climate change, and land use/land cover change (LUCC) plays a vital role in regulating climate system at different temporal and spatial scales. In this study, we analyzed the temporal and spatial trend of precipitation and the characteristics of LUCC on the QTP. Meanwhile, we also used the normalized difference vegetation index (NDVI) as an indicator of LUCC to discuss the relationship between LUCC and precipitation. The results show the following: (1) Annual precipitation showed a fluctuant upward trend at a rate of 11.5 mm/decade in this area from 1967 to 2016; three periods (i.e., 22 years, 12 years, and 2 years) of oscillations in annual precipitation were observed, in which expectant 22 years is the main oscillation period. It was predicted that QTP will still be in the stage of increasing precipitation. (2) The LUCC of the plateau changed apparently from 1980 to 2018. The area of grassland decreased by 9.47%, and the area of unused land increased by 7.25%. From the perspective of spatial distribution, the transfer of grassland to unused land occurred in the western part of the QTP, while the reverse transfer was mainly distributed in the northwestern part of the QTP. (3) NDVI in the northern and southwestern parts of the QTP is positively correlated with precipitation, while negative correlations are mainly distributed in the southeast of the QTP, including parts of Sichuan and Yunnan Province. Our results show that precipitation in the QTP has shown a fluctuating growth trend in recent years, and precipitation and NDVI are mainly positively correlated. Furthermore, we hope that this work can provide a theoretical basis for predicting regional hydrology, climate change, and LUCC research.

**Keywords:** Qinghai-Tibet Plateau; precipitation; trend analysis; wavelet analysis; ANUSPLIN; LUCC; NDVI

## 1. Introduction

Global climate changes, including change in regional precipitation pattern, has greatly influenced water circulation and hydrological processes [1]. Precipitation information is an important input for hydrological simulation, predicting of extreme precipitation events, and estimating the quantity and quality of surface water and groundwater [2]. It is an important natural water source for plants, especially in arid and semi-arid areas, and its changing trend has a great impact on plant growth and crop yields. The total amount of precipitation and time changes also have an important effect on soil water supply, water stress, and vegetation metabolism and physiological functions [3]. Therefore, understanding the temporal and spatial characteristics of precipitation is of practical significance in order to improve our understanding of precipitation variability analysis. In addition, affected by human activities such as deforestation, the expansion of urbanization, the expansion of agriculture, and the degradation of grassland, the natural quality of land cover has been seriously affected. The increasingly rapid development of land use/land cover change (LUCC) has begun to affect climate change and the sustainable utilization of resources.

Therefore, analyzing the relationship between LUCC and regional climate has become an important factor in understanding land–atmosphere interactions and designing climate adaptation and mitigation strategies [4].

Research reports show that China's annual precipitation has shown a certain upward trend since the middle of the 20th Century [5]. Affected by the East Asian monsoon, the climate change in China is characterized by "warm season precipitation" [6]. Under climate warming, precipitation in China has changed slightly, and regional heterogeneity has increased [7]. At present, researchers have proposed a variety of studies to explore precipitation changes over time in China [6,8,9]. Studies [10,11] have shown that precipitation in China increased by 2% between 1960 and 2000. From the perspective of seasonal distribution, the precipitation in winter and summer increased, while that in spring and autumn decreased over that period. At the same time, the regional distribution of precipitation showed a significant increase in northwest China (Qinghai Province, Xinjiang Uygur Autonomous Region, Gansu Province, etc.). During 1961–2015, the frequency and intensity of precipitation in northwest China had a unified upward trend [1]. In most cases, the precipitation data are from the measured data collected by meteorological stations, but it is difficult to describe the spatial distribution of precipitation due to the uneven distribution of the stations [1,2]. Therefore, it is necessary to use spatial interpolation to process observational data based on meteorological stations. Australian National University spline (ANUSPLIN) is a spatial interpolation program for meteorological data. This interpolation method is the integration and generalization of multiple linear regression, which has been implemented in many studies [1,9,12,13]. China has a vast territory, a wide range of latitudes, and different distances from the ocean. In addition, the terrain is different, and the types of landforms and mountain directions are diverse. As a result, the combination of temperature and precipitation is very different, forming a diverse climate in various places. Therefore, the distribution of precipitation in different regions has obvious inconsistent spatial and temporal trends [14,15].

With the development of the economy and the increase in population, the changes in land use have accelerated correspondingly, and the changes in land cover patterns have become increasingly obvious [16]. At present, LUCC has become one of the main problems affecting sustainable development and global environmental change [17–19]. LUCC also has a significant impact on regional and global climate change. Some of these effects are the result of direct impacts of LUCC on the local moisture and energy balance. Other impacts appear to be related to significant indirect climate impacts through the teleconnection processes [20,21]. LUCC mainly affects climate change in two ways: biogeochemical and biogeophysical processes. Biogeochemistry mainly affects climate change through the emissions of greenhouse gases, while biogeophysics mainly changes the water and heat transfer between the surface and the atmosphere due to changes in landscape and vegetation characteristics, thus impacting temperature and precipitation [22–24].

The Qinghai-Tibet Plateau (QTP) has an average altitude of more than 4000 m, a relatively special geographic location, and its topography is significantly higher than other areas in China. It is often called the "Roof of the World" or "Third Pole" [25,26]. It is an area that is sensitive to climate change in China and even the world [27,28]. In this study, we used ANUSPLIN interpolation to generate precipitation interpolation data with high spatiotemporal resolution in the QTP from 1967 to 2016. Based on the generated precipitation interpolation spatial data and the LUCC data we obtained, we synthetically analyzed the temporal and spatial characteristics of precipitation changes in the QTP and its relationship with LUCC. To provide references for the safety of water resources, the construction of the ecological environment on the QTP, the prediction of future climate change on the QTP, and the relationship between the characteristics of LUCC and precipitation are assessed.

## 2. Materials and Methods

### 2.1. Study Area

The QTP is located in southwestern China (Figure 1). Its geographical co-ordinates are between 26°00′ N–39°46′ N and 73°18′ E–104°46′ E, and it has an area of more than 2.5 million square kilometers, with an average elevation above 4000 m [25], thus making the region known as the "Roof of the World" [26]. The QTP includes 37 administrative cities in the Tibet Autonomous Region and Qinghai Province, the Xinjiang Uygur Autonomous Region, Gansu, Sichuan Province, and Yunnan Province [29]. It can be divided into six parts: Qiangtang Plateau, South-Tibet river basin, Tsaidam Basin, Qilian Mountains, Qinghai Plateau, and the Sichuan-Tibet Alpine Valley. Surrounded by mountains, interlaced valleys, and basins, the diverse topography results in a complex climate. The climate of the QTP is a unique plateau climate. About half of the entire QTP has an annual average temperature lower than 0 °C [30]. The average annual temperature of the QTP reduced from 20 °C in the southeast to below −6 °C in the northwest [31]. The southeastern region is the birthplace of many rivers such as the Yangtze River, the Yellow River, and the Lancang River, which contributed to the region being called the Asian water tower. The southeastern region is warm and humid, and the northwest region is cold and dry. The precipitation distribution is uneven. The annual precipitation in the southeastern Motuo area and the northwestern Lenghu area can reach 4000 and 17.6 mm, respectively. Under the influence of westerly circulation and plateau's topography, the average annual wind speed is greater than 3.0 m/s, and gale weather with wind speeds higher than 17 m/s occur on more than 50 days every year [32]. Due to its special geographical location and large-scale topography, the QTP has a strong influence on both regional and global climates [33].

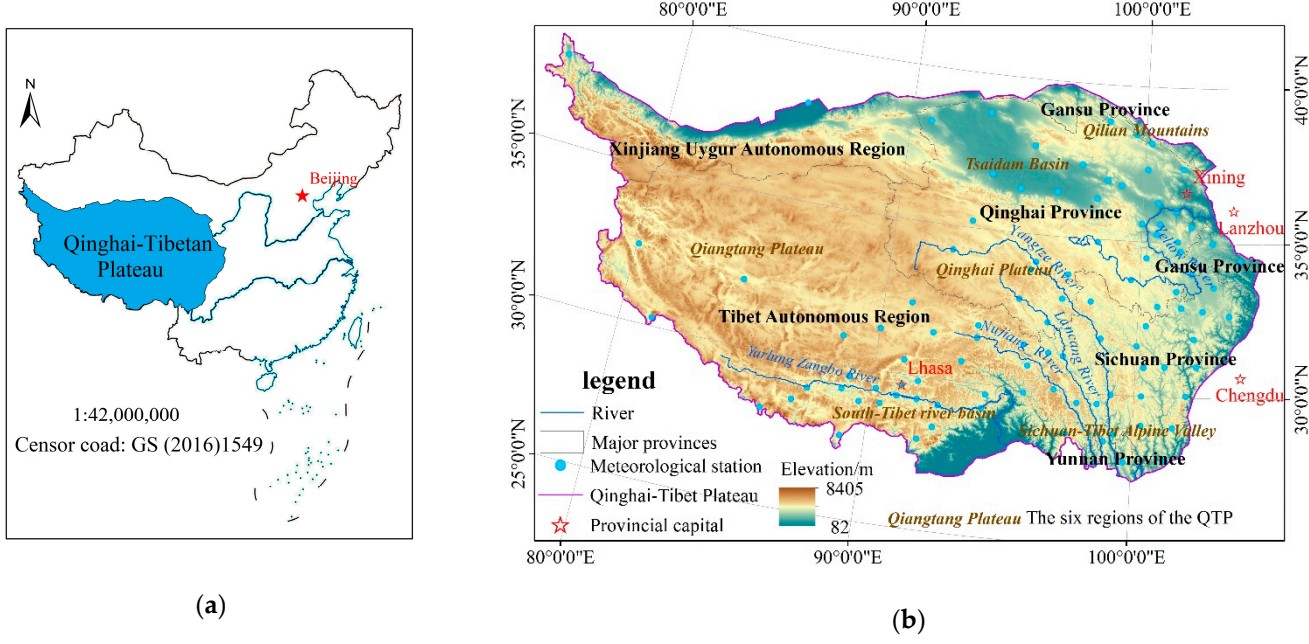

(**a**)                     (**b**)

**Figure 1.** Geographical location of the Qinghai-Tibet Plateau (QTP). (**a**) location of the QTP; (**b**) different regions of the QTP.

### 2.2. Data Resources and Processing

The precipitation data from 1967 to 2016 came from the daily dataset (V3.0) of the meteorological data of the China International Exchange Station of the National Meteorological Center (http://data.cma.cn/, accessed on 18 August 2020), which contains the monthly data of air pressure, temperature, precipitation, and other elements for 672 stations in China. Considering that the surrounding meteorological station data will have a certain impact on the interpolation of meteorological data in the QTP, the product data are used directly to interpolate the national precipitation data and are then based on the boundary

range of the QTP from the Data Center for Resources and Environmental Sciences, Chinese Academy of Sciences (RESDC) platform (http://www.resdc.cn/, accessed on 5 October 2020). With the help of ArcGIS10. 2, the platform performs batch cropping to obtain long-term precipitation spatial data on the QTP.

When discussing and analyzing the precipitation of the four seasons, the time according to the four seasons is divided into spring (March to May), summer (June to August), autumn (September to November), and winter (December to the following February) [5]. The winter data of 2016 were from December 2016 to February 2017. In order to reduce the seasonal variation in the precipitation series and make the data series conform to the characteristics of a stable random process, the anomaly value is selected in this paper to filter out the influence of monthly changes in the precipitation series. According to the relevant regulations of the World Meteorological Organization (WMO), the 30-year average precipitation from 1981 to 2010 is selected as the multi-year average to calculate the annual and seasonal average precipitation anomalies in the study area. A positive anomaly indicates that the precipitation value is higher than the 30-year average from 1981 to 2010, and a negative anomaly indicates that it is lower than the 30-year average.

The LUCC data of Qinghai, Tibet, Xinjiang, Yunnan, Sichuan, and Gansu provinces in 1980 and 2018 are also derived from the Resource and Environmental Science Data Center (RESDC) (http://www.resdc.cn/, accessed on 5 October 2020) [34] of the Chinese Academy of Sciences with a spatial resolution of 1000 * 1000 m. Chinese annual NDVI spatial distribution dataset is based on continuous time series of SPOT/VEGETATION NDVI satellite remote sensing data, using the maximum value composites (MVC) method to generate the annual NDVI dataset from 1998 to 2018 with a spatial resolution of 1000 * 1000 m. (http://www.resdc.cn, accessed on 5 October 2020) [35].

*2.3. Methodology*

This study mainly studies the temporal and spatial characteristics of precipitation in the QTP from 1967 to 2016 and the relationship between LUCC and precipitation. In this study, we have chosen the ANUSPLIN interpolation method to process precipitation data, because linear trend estimation and moving average method are the most basic and effective methods for analyzing the trend of time series data. Therefore, we use trend analysis to study the temporal change trend of the precipitation series on the QTP from 1967 to 2016. In the study of periodic changes in precipitation, since wavelet analysis is a widely used time-frequency analysis tool, it has the advantages of Fourier analysis. It can not only visually display the various periodic oscillations hidden in hydrological elements with time periods, but also display the time position of climate change. Additionally, wavelet analysis can objectively separate the data structure of different wavelengths, thereby displaying the wave amplitude on a graph. It is also a tool used to study the long-term changes in the power and amplitude of different meteorological variables [36–38]. Therefore, we use wavelet analysis to study the periodicity and future trend of changes and mutation points of the precipitation season series. In terms of spatial change characteristics research, the Theil–Sen trend analysis method named by Henri Theil and Pranab Sen is an effective linear trend detection method; it is not sensitive to outliers in a time series and is widely used in astronomical and environmental research [39–41]. The M-K test is a nonparametric test suitable for processing non-normally distributed data and has been widely used to describe the trends of climate and hydrological time series data [42,43]. This test is not sensitive to the interference of a few outliers and is especially effective for short-term time series data [44]. In this study, we combined the M-K test to determine the significance level of Theil–Sen analysis, and we used the Hurst index to detect the future trend of the precipitation series on the QTP. Finally, considering that the Pearson correlation coefficient is used to measure the degree of linear correlation between two variables and is the most widely used correlation coefficient, we chose this method to analyze the correlation between LUCC and precipitation.

2.3.1. Spatial Interpolation Method

At present, there are many studies on precipitation interpolation, but there are some differences in the processes and interpolation effects [1]. The choice of interpolation method depends on the purpose of the research and the environmental conditions of the area (e.g., the topography and distribution density of the precipitation observation stations) [45]. Traditional interpolation methods include the Thiessen polygon method [46,47], inverse distance weighted method (IDW) [48], and co-kriging interpolation (CK) [49]. These interpolation methods have been widely used in the interpolation of precipitation data [50]. While these interpolation methods are only suitable for application over comparatively flat areas, since they assume that the precipitation varies linearly between stations, they do not consider the impact of terrain on precipitation, which is one of the most significant factors influencing the distribution of precipitation [1]. However, the terrain in China is complex and varied, and precipitation is greatly affected by the terrain. The Australian National University spline (ANUSPLIN [51]) interpolation method can effectively address the impact of terrain on precipitation. The method based on thin plate smoothing splines technology has been widely used [52,53], and it has been proven that the results obtained in precipitation interpolation are reliable, and the interpolation results are better than those from Kriging interpolation [12]. The partial thin-plate smoothing spline model for the predicted value $Z_i$ at location $x_i$ is shown as follows [1]:

$$Zi = f(x_i) + \sum_{j=1}^{p} \beta_j \Psi_j(x_i) + \varepsilon_i \ (i = 1, \ldots, n; j = 1, \ldots, p) \tag{1}$$

In Equation (1), n is the number of observational data, f represents a smoothing function which needs to be estimated, $\beta_j$ denotes a series of parameters (p dimensions), which also needs to be estimated, $\Psi_j$ are a series of function (p dimensions) of independent variables, and the $\varepsilon_i$ represent independent, random, and zero mean errors.

In this study, the digital elevation model (DEM) elevation data (data source: http://www.resdc.cn/, accessed on 5 October 2020) with a spatial resolution of 1000 * 1000 m is used as a covariate when we interpolate the data, so the final processing results in the precipitation raster data with a spatial resolution of 1000 * 1000 m.

2.3.2. Trend and Variation Analysis

Linear tendency estimation uses linear regression to solve the correlation change in the signal sequence with respect to the time change to determine the data change trend. The moving average law uses low-pass filtering to determine the trend. Anomaly analysis can intuitively detect mutation points in the sequence through changes in anomalies.

Theil–Sen trend estimation. To determine the spatial change in the precipitation series in the past 50 years, the Theil–Sen trend was used to analyze the precipitation time series in the QTP. The magnitude of the slope can be calculated according to Equation (2) [54]:

$$Qsen = median \frac{X_j - X_i}{j - i} \ 1 < i < j < n \tag{2}$$

In Equation (2), the Sen slope is represented by $Q_{sen}$, where $X_i$ and $X_j$ are the observation data corresponding to time points i and j, respectively. If the time series consists entirely of n observations, there will be $(n(n-1))/2$ estimated slopes. $Q_{sen}$ is the result of the statistical test, and the median of the estimated slopes is taken. In the analysis, a $Q_{sen}$ greater than 0 indicates that the sequence has an upward trend, while a $Q_{sen}$ less than 0 indicates that the sequence has a downward trend [55].

Hurst exponent and detection of future precipitation trends. The Hurst index value varies from 0 to 1. A value between 0.5 and 1 indicates that the time series data are persistent. When the Hurst index value is close to 1, the persistence characteristics in the time series will become more prominent. A Hurst index value less than 0.5 indicates a time series with anti-persistence, indicating that the development trend of the future time

series may change [56]. We chose R/S rescaled range analysis based on the Hurst index to measure the long-term dependence of precipitation time series [57]. R/S analysis is used to estimate the autocorrelation characteristics of a time series [58]. Mainly by defining the ratio of the range to the standard deviation as R/S, the following exponential law can be obtained:

$$R/S = C * t^H \tag{3}$$

In Equation (3), R is the range sequence; S is the standard deviation sequence; c is a constant; t is the length of the time sequence. The Hurst exponent can be obtained by the least multiplication method on the upper logarithmic graph of t and R/S according to the measured data [59].

### 2.3.3. Wavelet Analysis

For the time series function f(t), the wavelet transform is defined as follows [60,61]:

$$W_f(a,b) = |a|^{-\frac{1}{2}} \int_{-\infty}^{+\infty} f(t)\varphi^*\left(\frac{t-b}{a}\right) dt \tag{4}$$

In Equation (4), $W_f(a,b)$ is the wavelet coefficient; a is the expansion factor; b is the translation factor; t is the time; f(t) is the arbitrary square integrable function, that is, the precipitation process; $\varphi(t)$ is a basic wavelet (Mother wavelet); $\varphi^*$ is the conjugate function of $\varphi$. Using this function to draw the contour map of wavelet coefficients, the periodic characteristics and sudden changes in precipitation series can be identified.

The Morlet function is a complex wavelet with good time-frequency locality. Its function is defined as follows:

$$\varphi(t) = \exp\left(i\omega_0 t - \frac{t^2}{2}\right) \tag{5}$$

In Equation (5), $\omega_0$ is a dimensionless frequency. When $\omega_0 \geq 5$, the Morlet wavelet can approximately satisfy the allowable condition. This paper chose this wavelet to perform discrete wavelet analysis of precipitation series based on the MATLAB platform.

The wavelet variance is the integration of the square of all wavelet variation coefficients of the relevant year on the time scale, and the expression is as follows [60]:

$$Var(a) = \int_{-\infty}^{+\infty} |W_f(a,b)|^2 db \tag{6}$$

In Equation (6), $Var(a)$ is the wavelet variance, and $W_f(a,b)$ is the wavelet coefficient. It can be used to draw a wavelet variance graph, which can be used to determine the oscillation period of the precipitation sequence.

### 2.3.4. Correlation Analysis

The calculation formula is as follows:

$$r = \frac{\sum_{i=1}^{n}(x_i - \bar{x})(y_i - \bar{y})}{\sqrt{\sum_{i=1}^{n}(x_i - \bar{x})^2 \sum_{i=1}^{n}(y_i - \bar{y})^2}} \tag{7}$$

In Equation (7), $x_i$ and $y_i$, respectively, represent the values of two variables; $\bar{x}$ and $\bar{y}$ are the average values of the two variables, respectively; *r* is the correlation coefficient, and when *r* > 0, the two variables are positively correlated, while when *r* < 0, the two variables are negatively correlated. A larger the |*r*| indicates a stronger correlation.

### 3. Results

*3.1. Characteristics of Precipitation on the QTP*

3.1.1. Analysis of Temporal Variation Trend of Precipitation

From 1967 to 2016, the overall change in precipitation on the QTP showed an increasing trend (Figure 2a). Trends of precipitation was 11.5 mm/decade, and the 10-year moving average showed fluctuating growth. According to the change trend of the anomaly of annual precipitation (Figure 2b), precipitation fluctuated and increased before the 1990s, and the anomaly exhibited alternating positive and negative changes. From the early 1990s to the early 2000s, the precipitation experienced decreasing fluctuations, and the anomaly value was mainly negative. After entering the 21st century, the precipitation was in a significant increasing stage, the anomaly value was mainly positive. The change rates of climate were 4.1 mm/decade (Figure 2c), 6.3 mm/decade (Figure 2d), 0.3 mm/decade (Figure 2e), and 0.6 mm/decade (Figure 2f) for the spring, summer, autumn, and winter, respectively. Overall, the change trend of the four seasons was partly the same as that of the whole year, but the precipitation tendency rate in spring and summer was higher than that in the other two seasons, indicating that the precipitation change trend in spring and summer was obvious.

From the perspective of the 10-year moving average trend, the change trend of the spring and autumn seasons was similar to that of the whole year, particularly in spring. The change trends in winter and summer showed that from the early 1990s to the beginning of the 21st century, the declining trend was not obvious, and it was still in the volatile growth stage; however, the performance of the other two stages was still relatively obvious, especially in summer. The change trend was basically the same as the change throughout the whole year. In addition, during the three seasons of spring, autumn, and winter, precipitation began to decrease in the 2010s, but that in summer still showed a clear increasing trend, indicating that the trend of precipitation in the late 21st century indicated precipitation was mainly concentrated in summer.

Table 1 shows that the annual and seasonal average precipitation before the 1990s showed fluctuating changes, and the results from 50 years show that it is still in a fluctuating growth trend. The average value in the 1990s was lower than other years (Table 1), and it can be proved by comparing with the results in Figure 2. The average value of the autumn and winter seasons in 2010–2016 was lower than the value in the early years of the 21st century but higher than the value in the 1990s, indicating that the growth of autumn and winter precipitation has slowed. The average values of the spring and summer from 2010 to 2016 and 50 years were higher than those in the early 2010s, indicating that precipitation is still increasing, but it is mainly concentrated in spring and summer, especially summer.

**Table 1.** The average value of precipitation indifferent ages in the QTP (mm).

|  | 1967–2016 | 1967–1969 | 1970–1979 | 1980–1989 | 1990–1999 | 2000–2010 | 2010–2016 |
|---|---|---|---|---|---|---|---|
| Annual | 413.4 | 397.5 | 400.9 | 405.2 | 388.0 | 436.9 | 447.0 |
| Spring | 69.3 | 60.2 | 66.4 | 66.9 | 58.5 | 77.9 | 83.8 |
| Summer | 250.9 | 247.8 | 240.7 | 245.3 | 242.7 | 261.3 | 271.8 |
| Autumn | 81.1 | 79.1 | 82.2 | 81.5 | 76.2 | 84.7 | 81.8 |
| Winter | 12.1 | 11.1 | 11.2 | 11.8 | 10.7 | 15.2 | 12.2 |

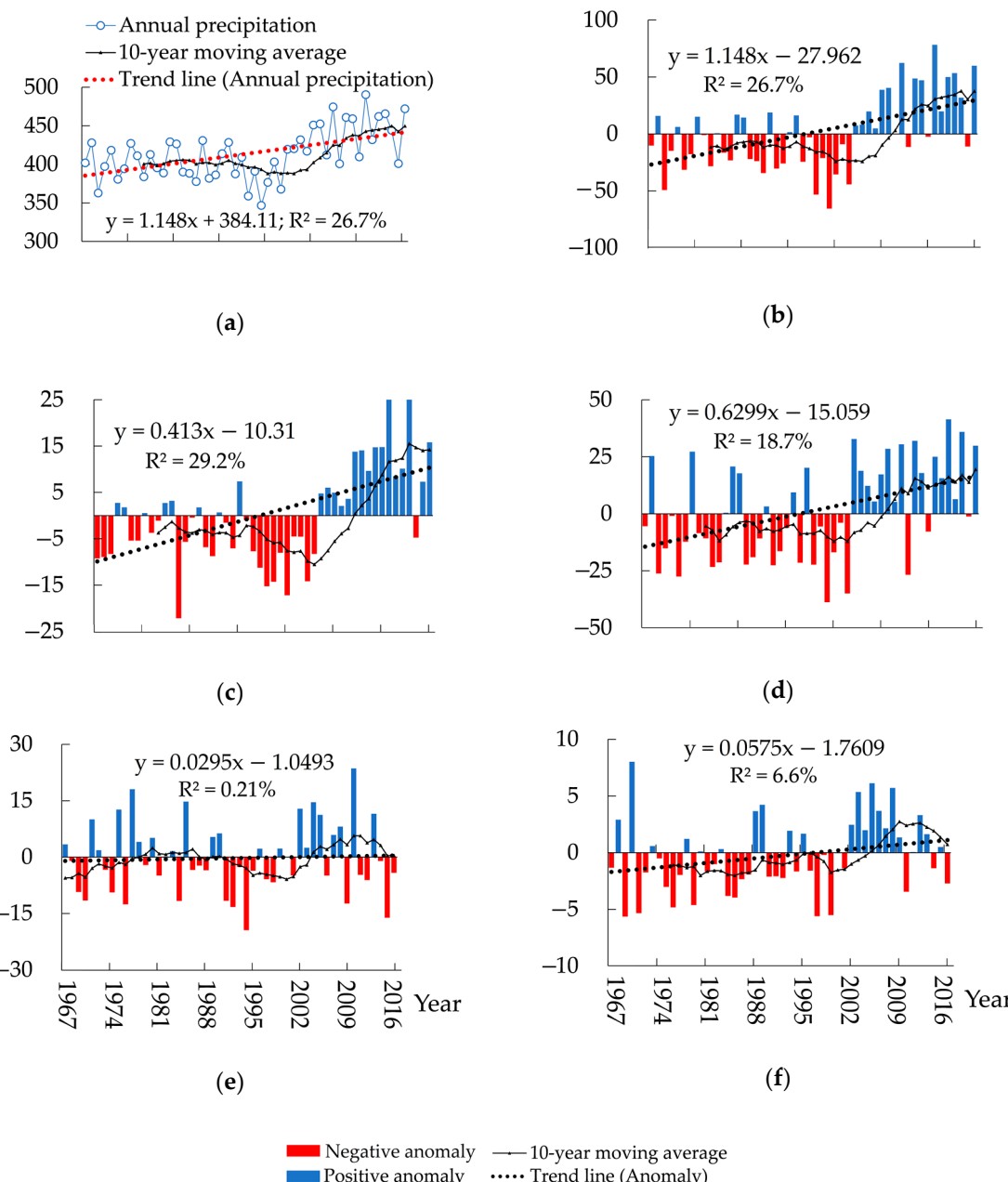

**Figure 2.** Variation of annual precipitation on the QTP during 1967–2016, the changes in annual and seasonal precipitation anomalies on the QTP. (**a**) Variation in annual precipitation; (**b**) annual precipitation anomaly; (**c**) spring precipitation anomaly; (**d**) summer precipitation anomaly; (**e**) autumn precipitation anomaly; (**f**) winter precipitation anomaly.

### 3.1.2. Periodicity Analysis of Precipitation Series

The precipitation on the QTP from 1967 to 2016 had a significant change cycle over 50 years (Figure 3). The wavelet variance diagram of annual precipitation over 50 years (Figure 3a) showed that there were three obvious oscillation period (i.e., 2, 12, and 22 years). Plotting the change process of the real part of the wavelet coefficients of 22 and 12-year cycles (Figure 3b), we can clearly see the precipitation fluctuations. Blue is the 22-year scale, and it has experienced five high and low changes. Table 2 shows that the periods with low precipitation are 1967~1973, 1984~1995, and 2006~2016. The periods with heavy precipitation were 1973~1984 and 1995~2006. According to the data analysis in Figure 3b, 2016 is in the low water period of precipitation. From the analysis of the two change cycles,

it is estimated that 2017 represented a transition point from less to more precipitation, and future precipitation is expected to show an increasing trend.

Similar to the annual precipitation trend, the wavelet variance map of spring precipitation (Figure 3c) shows that there were three obvious oscillation period (i.e., 4, 12, and 22 years). Figure 3d and Table 2 show that the 22 years cycle of spring precipitation also experienced 5 high and low changes. According to the 22-year cycle, it is expected that the average spring precipitation will also show an increasing trend in the future. Figure 3e showed that there were three obvious oscillation period (i.e., 4, 10, and 24 years) (summer precipitation). Combining Figure 3f and Table 2 shows that the 24-year cycle has also experienced five rich and withered changes. From the data analysis in the figure, 2019 is the time point at which precipitation transitioned from less to more, and it is expected that future precipitation will show an increasing trend. The wavelet variance map of autumn precipitation (Figure 3g) shows that there were three obvious oscillation period (i.e., 2, 12, and 24 years). Figure 3h and Table 2 show that the 24 years cycle of autumn precipitation also experienced five high and low changes. According to the data analysis in the figure, 2016 was in the low water period of precipitation, and the overall precipitation amount was relatively small. From the analysis of the 24 years trend line, 2019 was the time point at which precipitation transitioned from less to more. Therefore, future precipitation is expected to show an increasing trend. The wavelet variance map of winter precipitation (Figure 3i) shows that there were three obvious oscillation period (i.e., 2, 12, and 20 years). The 24-year cycle of winter precipitation also experienced six high and low changes (Figure 3j). From the data analysis in the figure, 2016 was in a rainy season, and 2021 is expected to be a transition point at which precipitation changes from more to less. Therefore, future precipitation is expected to show a decreasing trend.

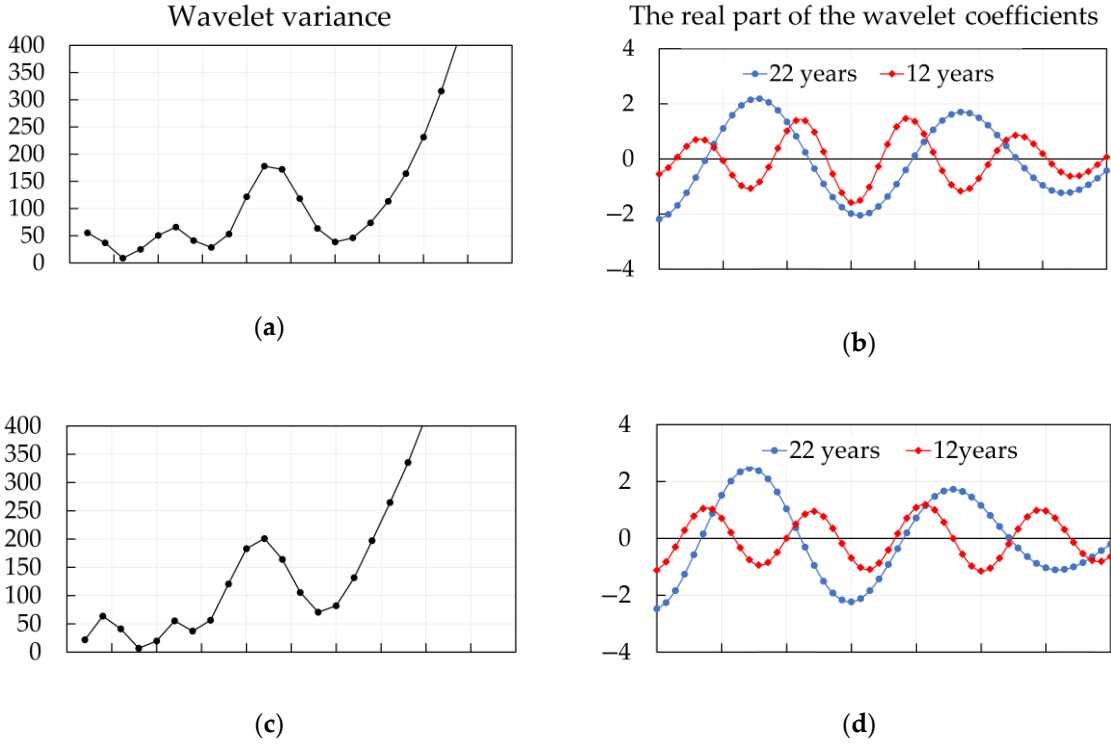

**Figure 3.** *Cont*.

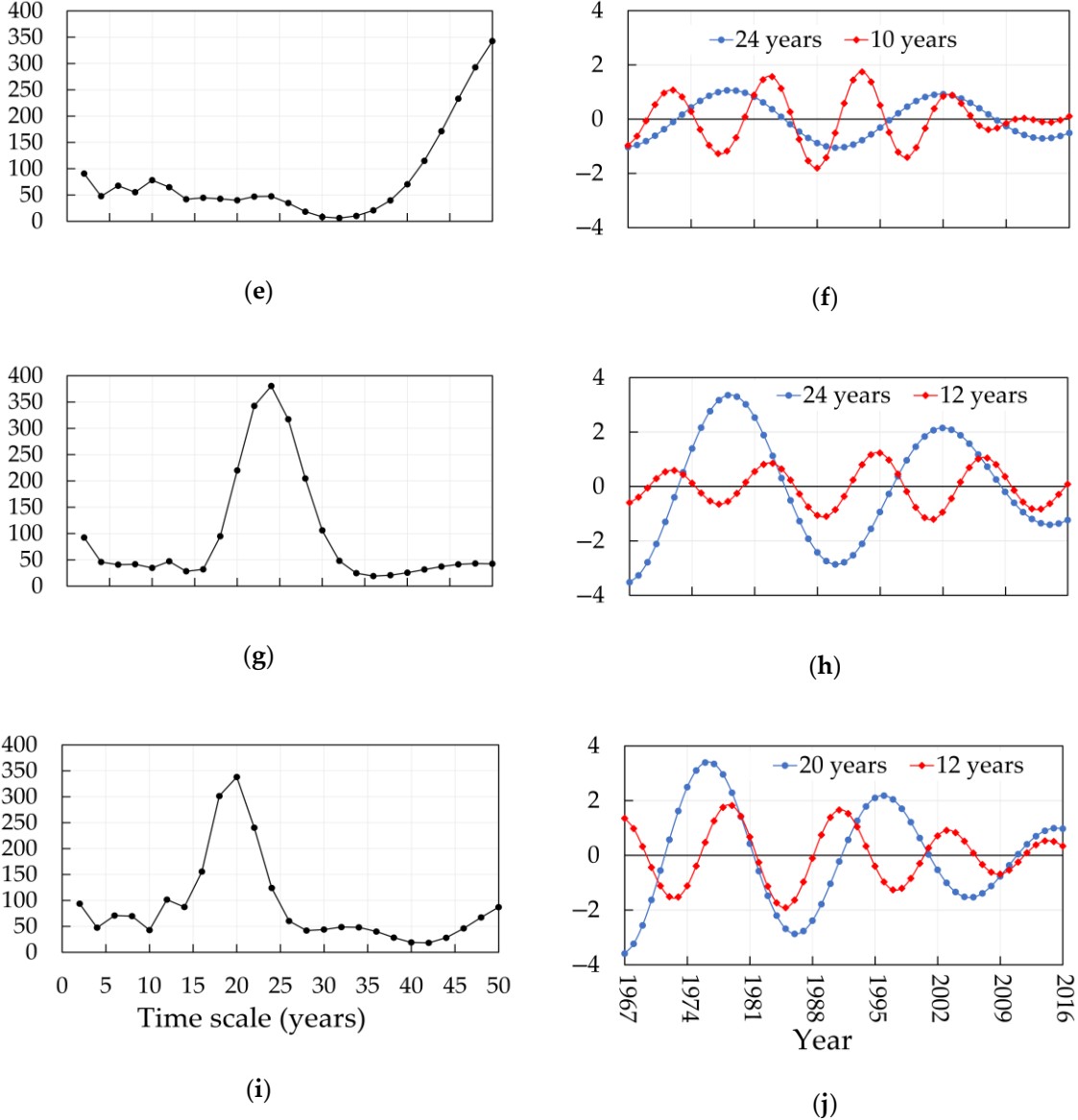

**Figure 3.** Wavelet variance plot of the annual average precipitation series and the seasonal average precipitation series on the QTP: (**a**) annual; (**c**) spring; (**e**) summer; (**g**) autumn; (**i**) winter. The real part of the wavelet coefficients of the annual average precipitation series and seasonal average precipitation series on the QTP: (**b**) annual; (**d**) spring; (**f**) summer; (**h**) autumn; (**j**) winter.

**Table 2.** Disruption points of real value of wavelet coefficient of annual and seasonal precipitation in QTP.

| | Oscillation Period (Years) | Precipitation Change Node | | Infer the Change Node and Duration |
|---|---|---|---|---|
| | | From Abundant to Dry | From Dry to Abundant | |
| Annual | 22 | 1984, 2006 | 1973, 1995 | 2017 from dry to abundant, 2017–2028 |
| | 12 | 1973, 1985, 1997, 2009 | 1968, 1979, 1991, 2003, 2016 | 2016–2021 |
| Spring | 22 | 1983, 2006 | 1972, 1994 | 2017 from dry to abundant, 2017–2028 |
| | 12 | 1975, 1986, 1999, 2012 | 1969, 1980, 1992, 2006 | 2017 from dry to abundant, 2017–2022 |
| Summer | 24 | 1985, 2008 | 1973, 1996 | 2019 from dry to abundant, 2019–2030 |
| | 10 | 1974, 1985, 1996, 2006 | 1969, 1980, 1991, 2001, 2016 | 2016–2021 |
| Autumn | 24 | 1985, 2008 | 1973, 1996 | 2019 from dry to abundant, 2019–2030 |
| | 12 | 1974, 1985, 1997, 2008 | 1969, 1979, 1991, 2003, 2016 | 2016–2021 |
| Winter | 20 | 1981, 2001 | 1972, 1991, 2011 | 2021 from abundant to dry, 2021–2031 |
| | 12 | 1970, 1981, 1995, 2007 | 1975, 1989, 2001, 2011 | 2017 from abundant to dry, 2017–2022 |

### 3.1.3. Analysis of Spatial Variation Trend of Precipitation

Through ANUSPLIN interpolation and ArcGIS processing, the spatial distribution of the 50-year seasonal precipitation on the QTP was obtained (Figure 4). Figure 4e shows that the spatial distribution of precipitation on the QTP is significantly different, and the precipitation decreased from the southeast to northwest. The southeast is the source of China's main rivers, indicating that the southeast has sufficient precipitation. The precipitation in the northwest showed a decreasing trend as the elevation increased. From the perspective of the spatial distribution of seasonal precipitation, the spatial distribution trend of average precipitation in spring (Figure 4a), summer (Figure 4b), and autumn (Figure 4c) was similar to the spatial distribution of annual precipitation (Figure 4e), especially in summer. They all appeared to decrease from the southeast to northwest, and the winter (Figure 4d) had the lowest precipitation, which was below 200 mm overall, with small spatial differences.

Figure 5e shows that the spatial variation in precipitation throughout the year from 1967 to 2016 showed an increasing trend, with the highest growth slope reaching 17.1 mm/year. Except for the 19.25% of areas in the southern and eastern parts of the QTP, which decreased significantly, most of the remaining areas showed an increasing trend. The growth trend in high-elevation areas was relatively obvious. In terms of seasonal growth trends, the spatial growth trends in spring (Figure 5a) and winter (Figure 5d) were lower than those in summer (Figure 5b) and autumn (Figure 5c). The spatial growth in summer and autumn was relatively obvious. The slope of change was positive, showing an increasing trend. However, the precipitation in spring increased significantly in the southern part of the QTP, with a slope of 9.2 mm/year. The slope of change in winter had a lower value for each seasonal variation, with the highest slope being 1.8 mm/year. The calculation results of the H showed that the indexes for the whole year and the four seasons of spring, summer, autumn, and winter at 50 years were 0.79, 0.89, 0.75, 0.71, and 0.81, respectively. The results were all greater than 0.5, indicating that the current changes had obvious persistence and that precipitation will likely show an increasing trend.

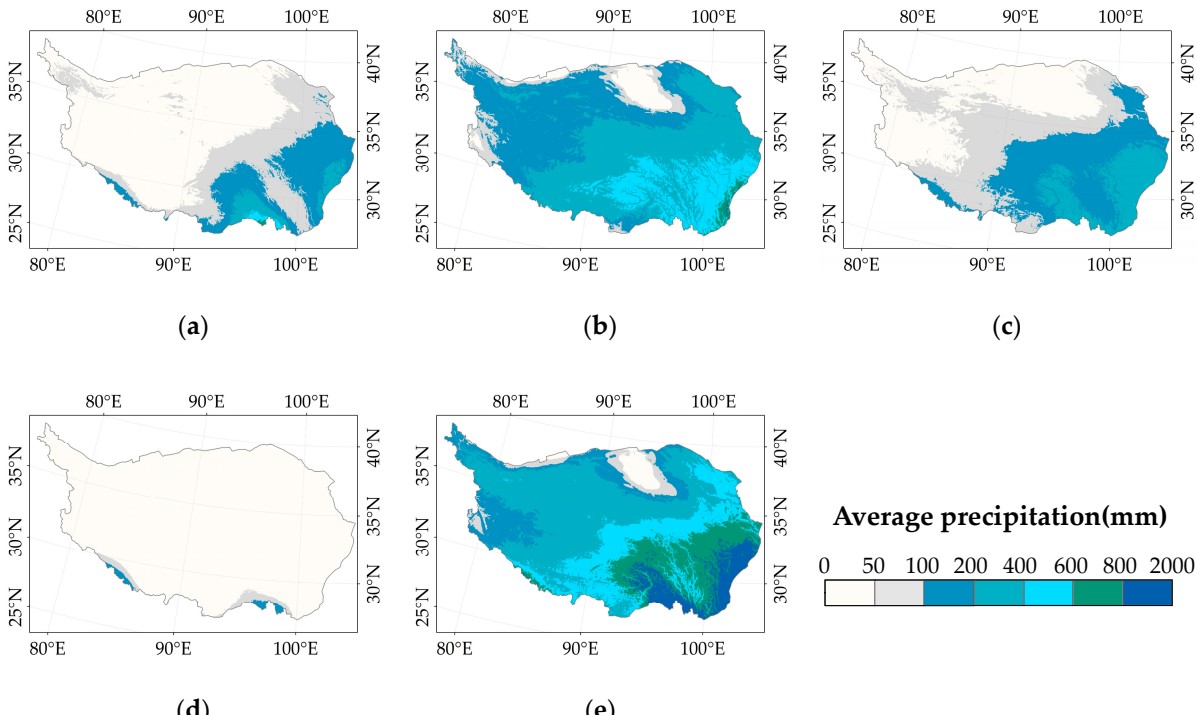

**Figure 4.** Spatial distribution of average precipitation from 1967 to 2016: (**a**) spring; (**b**) summer; (**c**) autumn; (**d**) winter; (**e**) annual.

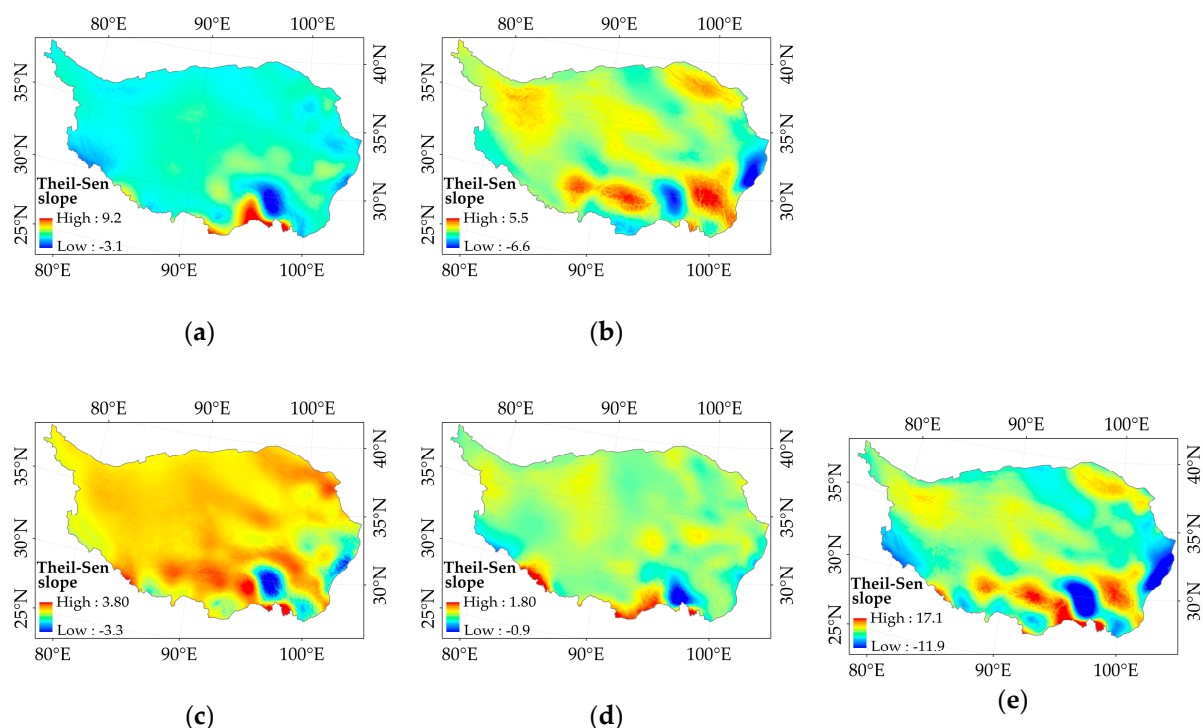

**Figure 5.** Theil–Sen slope of precipitation on the QTP from 1967 to 2016: (**a**) spring; (**b**) summer; (**c**) autumn; (**d**) winter; (**e**) annual.

### 3.2. Characteristics of LUCC on the QTP

Based on the land-use type data of the five provinces of Qinghai, Tibet, Xinjiang, Yunnan, and Sichuan in 1980 and 2018 and the boundary of the QTP, the spatial distribution map of the land-use types of the QTP in 1980 and 2018 (Figure 6) was obtained.

Figure 6 and Table 3 show that the land-use structure of the QTP underwent significant changes from 1980 to 2018. The proportions of grassland and unused land changed significantly. Grassland accounted for 58.17% of the total in 1980 and declined to 48.70% in 2018, a decrease of 9.47%. The proportion of unused land increased from 25.86% in 1980 to 33.11% in 2018, an increase of 7.25%. Grassland and unused land are mainly distributed on the western QTP and the northwestern Tibet Autonomous Region. Figure 6e shows that the precipitation growth trend in this area was not obvious, and the precipitation growth in some areas even showed a decreasing trend; thus, the reduction in precipitation may more drastically shift the grassland to unused land.

Combining the main spatial transfer map of land-use types (Figure 7) and the proportion of the transfer matrix results (Table 4), the proportion of land transferred from grassland to unused land reached 15.18%, which was the largest proportion of land transfer. The distribution of this type of transfer is obvious in Figure 7, mainly in the west of the QTP, that is, the northwestern Tibetan Autonomous Region, indicating that there is a certain grassland degradation phenomenon in this area. The transfer area from unused land to grassland accounted for 7.61%, indicating that the area of transferred grassland was much larger than the transfer area. This transfer type was widely distributed in the northwestern part of the QTP, namely, the southern part of the Xinjiang Uygur Autonomous Region, the southern part of the QTP, and the eastern part of the QTP. That is, there were obvious distributions in the areas of Gansu Province in Qinghai Province. The proportion of land transferred from woodland to grassland reached 3.16% (Table 4) and was mainly distributed in the southern part of the QTP, while the eastern part was relatively scattered. The transfer from grassland to woodland accounted for 4.30%, mainly in the southern part of the Tibetan Plateau, including southeastern Tibet and Gansu. Southwestern and northwestern Sichuan are areas where major rivers flow through, and precipitation is relatively abundant, resulting in the phenomenon that some grasslands were transferred to forestland during 1980–2018. The rest of the transfer distribution was relatively scattered and evenly distributed in all areas.

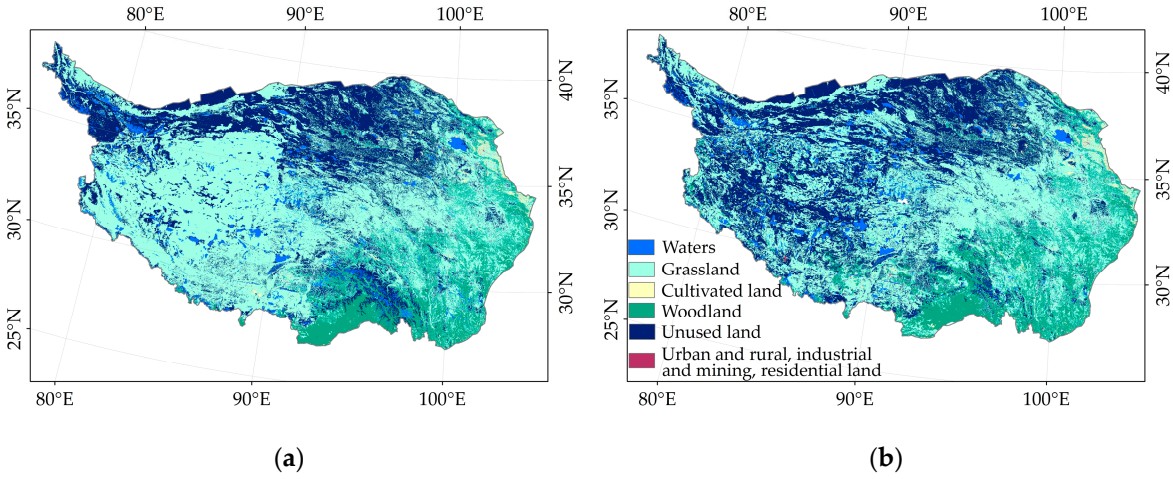

**(a)** **(b)**

**Figure 6.** Land use and cover changes on the QTP: (**a**) 1980; (**b**) 2018.

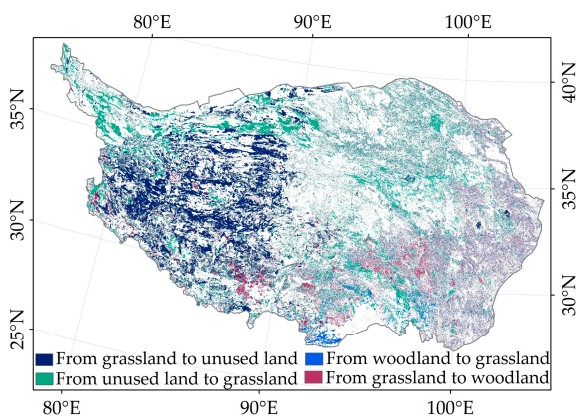

**Figure 7.** The main spatial transfer of land-use types from 1980 to 2018.

**Table 3.** Statistics of land-use changes on the QTP from 1980 to 2018.

| Code | Land-Use Type | Area Ratio/% | | 1980–2018 Change |
|---|---|---|---|---|
| | | **1980** | **2018** | |
| 1 | Cultivated land | 0.89 | 1.03 | 0.14% |
| 2 | Woodland | 10.42 | 12.04 | 1.62% |
| 3 | Grassland | 58.16 | 48.69 | −9.47% |
| 4 | Waters | 4.62 | 5.03 | 0.41% |
| 5 | Urban and rural, industrial, mining, residential land | 0.05 | 0.10 | 0.05% |
| 6 | Unused land | 25.86 | 33.11 | 7.25% |

**Table 4.** The proportion of land-use type spatial transfer area on the QTP from 1980 to 2018 (%).

| | | 2018 | | | | | | |
|---|---|---|---|---|---|---|---|---|
| | **Land-Use Type Code** | **1** | **2** | **3** | **4** | **5** | **6** | **Proportion Total** |
| | **1** | 0.35 | 0.15 | 0.31 | 0.03 | 0.03 | 0.02 | 0.89 |
| | **2** | 0.18 | 6.75 | 3.16 | 0.07 | 0.01 | 0.25 | 10.42 |
| **1980** | **3** | 0.43 | 4.30 | 36.64 | 1.57 | 0.04 | 15.18 | 58.16 |
| | **4** | 0.02 | 0.12 | 0.95 | 2.44 | 0.00 | 1.09 | 4.62 |
| | **5** | 0.02 | 0.00 | 0.02 | 0.00 | 0.01 | 0.00 | 0.05 |
| | **6** | 0.03 | 0.72 | 7.61 | 0.92 | 0.01 | 16.57 | 25.86 |
| | **Proportion total** | 1.03 | 12.04 | 48.69 | 5.03 | 0.10 | 33.11 | 100.00 |

*3.3. Relationship between Precipitation Change and LUCC*

There are many factors that affect the NDVI [17], and LUCC can greatly affect the changes in the NDVI [62]. Therefore, the following results may explain only part of the impact of LUCC because the NDVI is used as an indicator of LUCC in this study [62–64]. During the period from 1998 to 2018 on the QTP, the NDVI value presented an obvious linear growth trend with a growth rate of $2.4 \times 10^{-3}$ per year (Figure 8). The overall average NDVI distribution was low in the northwest and high in the southeast, gradually decreasing from the northwest to southeast (Figure 9a). Figure 9b shows that areas with decrease in the NDVI (Theil–Sen slope < 0) accounted for 31.69%, which were mainly distributed in the western and northern regions of the QTP. While areas with increase in the NDVI (Theil–Sen slope > 0) accounted for 68.31%, which was mainly distributed in the eastern and southern parts of the Tibetan Plateau. Areas with a significant decrease in the NDVI ($p < 0.05$, Theil–Sen slope < 0) accounted for 6.74% (Table 5), which were more clearly distributed in north Tibet, the alpine valleys in the southwestern QTP and the Nu River in the southern parts of the plateau. A total of 50.17% of the NDVI-increasing trend passed the significance test, mainly due to a significant increase of 45.15% ($p < 0.05$) and a

slight increase (0.05 < *p* < 0.1) of 5.02% (Table 5), which were widely distributed throughout the QTP and mainly concentrated in the east and south.

The correlation between the precipitation and NDVI was calculated from 1998 to 2016. According to the significance test of the correlation coefficient, at a significance level of 0.05, the critical value was 0.456, and at a significance level of 0.01, the critical value was 0.575. Figure 9d showed that the positive correlation accounted for 68.35%, which was widely distributed throughout the QTP, indicating that precipitation was generally beneficial to the growth of vegetation; additionally, the significant positive correlation (critical value > 0.575, *p* < 0.01) accounted for a proportion of 4.59%, mainly distributed in the northeastern part of the QTP, including parts of Gansu and Qinghai and parts of Xinjiang in the northwest. The generally significant positive correlations (0.456 < critical value < 0.575, 0.01 < *p* < 0.05) accounted for 7.10%, which were mainly distributed in the northeast and southwest of the plateau. Negative correlations accounted for 31.65% and were mainly distributed in the western and southern parts of the QTP; however, the proportions that passed the significance test were relatively small. The critical value was less than −0.456 (*p* < 0.05), accounting for 1.21%, mainly distributed in the southeast of the plateau, including parts of Sichuan and Yunnan.

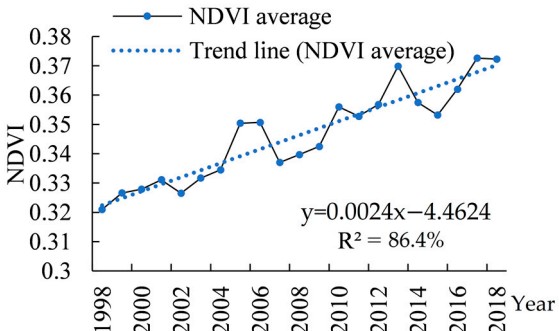

**Figure 8.** Time change trend of the normalized difference vegetation index (NDVI) from 1998 to 2018.

**Table 5.** The proportion of the NDVI and its change trend from 1998 to 2018.

| NDVI Value Classification | Proportion | NDVI Change Trend | Proportion |
|---|---|---|---|
| [0,0.15) | 32.06% | Slightly reduced | 2.99% |
| [0.15,0.3) | 24.81% | Significantly reduced | 6.74% |
| [0.3,0.5) | 12.57% | stable | 40.10% |
| [0.5,0.7) | 14.06% | Slight increase | 5.02% |
| [0.7,0.90] | 16.50% | Significantly increased | 45.15% |

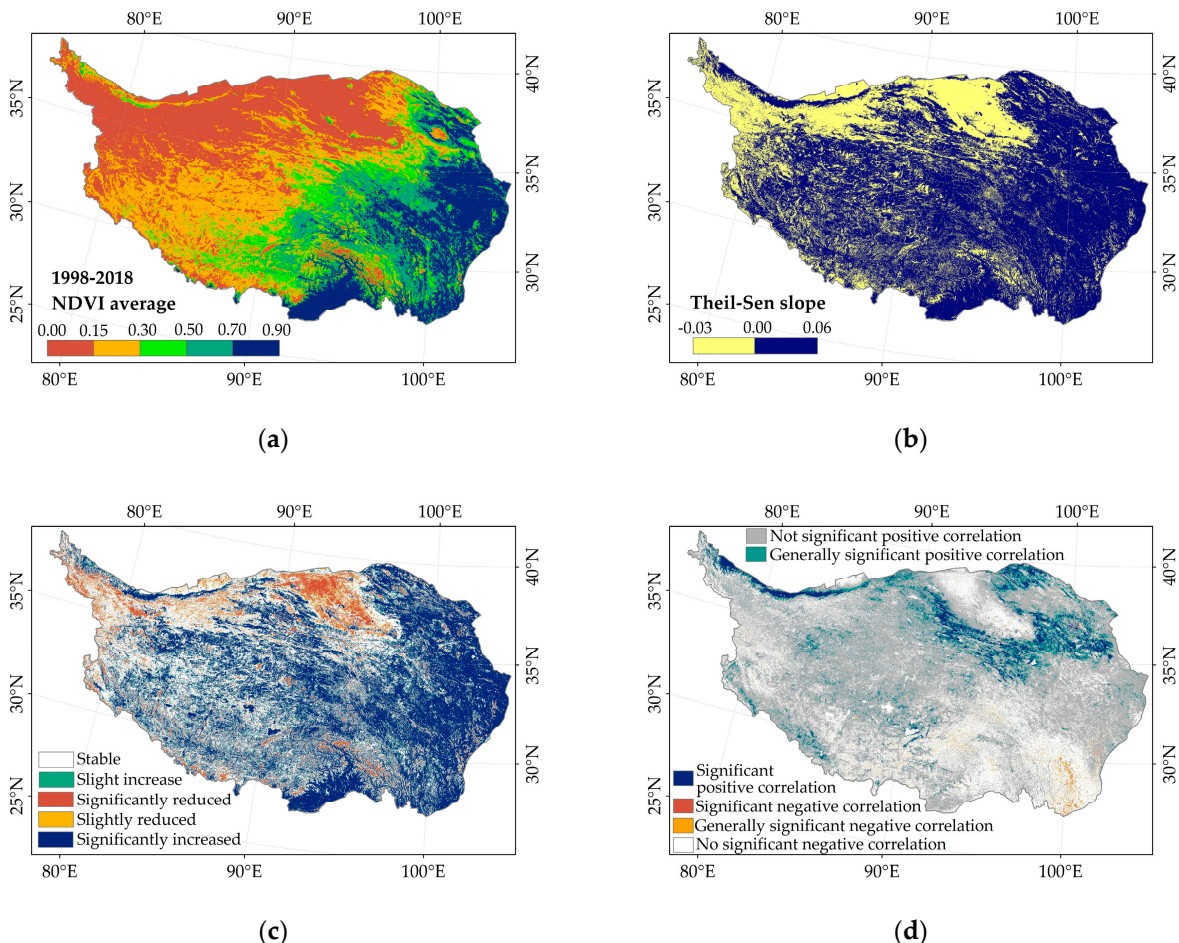

**Figure 9.** The spatial distribution and change trend of the NDVI from 1998 to 2018 and its correlation with precipitation: (**a**) average value; (**b**) Theil–Sen slope; (**c**) change trend; (**d**) 1998–2016 correlation between the NDVI and precipitation.

## 4. Discussion

This study presents the long-term (1967 to 2016) spatial–temporal characteristics of precipitation on the QTP with precipitation interpolated by ANUSPLIN. Precipitation is greatly affected by topography and the terrain on the QTP is diverse. The ANUSPLIN interpolation method considering the influence of terrain can produce reliable precipitation interpolation data. Comparing to conventional interpolation method, the ANUSPLIN show its priority in modelling the precipitation under complex terrain and data-scarce condition. Although the general spatial distributions of precipitation generated from ANUSPLIN interpolation and conventional interpolation output are similar in China, the ANUSPLIN precipitation grid gives much more accurate distribution information for precipitation [1]. For example, the daily gridded precipitation dataset (with a resolution of 0.5°) developed by the China Meteorological Administration is also processed using the ANUSPLIN interpolation method [65].

In this study, we analyzed the annual and seasonal variation cycles of precipitation on the QTP using wavelet analysis. Previous studies analyzed the variation cycles of precipitation in various parts of China. For example, the precipitation cycle in Yunnan Province [66] from 1954 to 2012 was mainly concentrated at 22 years and 10 years, while in winter, it was 4, 9, and 17 years, and in autumn, it was 3, 10, and 18 years. In Tibet [67], it was concentrated in 4, 10, and 20 years. The change cycle in Xinjiang from 1961 to 2017 was concentrated in three cycles: 10–13, 22–28, and 44–50 years [68]. These results shows that the change cycle of precipitation is characterized by the superimposition of the three cycles of large–medium–small precipitation. The cycle scales obtained by selecting different

research areas and research periods may have corresponding differences, but in the case of small differences in spatial locations, there were basically similar periodic scales, and most of them appeared as the three change cycles of large–medium–small precipitation. The factor that affects the climate cycle may be the role of the solar radiation, cosmic geophysical factors, atmospheric circulation [69,70]; changes in the orbital parameters of the Earth due to its movement around the Sun [71]; changes in the intensity of galactic cosmic rays [72]; the strong impact of human activity [73]. From the analysis of the spatial change trend of precipitation, we found that most areas of the QTP showed an increasing trend, except for the 19.25% of areas in the east, west, and south. This conclusion can be verified in the following article [1,5,10,11]. At present, precipitation in most parts of China shows an increasing trend [10,11], while the frequency and intensity of precipitation in the northwestern region have a unified upward trend [1]. The spatial growth trend of precipitation is obvious in Qinghai Province, mainly in high-elevation areas such as the southwest and northeast of the region. This conclusion is verified in some studies on precipitation characteristics in Qinghai Province [5]. From the perspective of seasonal change trends, this study concludes that the increasing trend in summer and autumn was obvious, especially in summer, and it was consistent with the conclusions in related research in which the precipitation in winter and summer increased and the precipitation in spring and autumn decreased [11].

In this work, characteristics of LUCC on the QTP showed that there is grassland degradation on the western QTP, that is, the land type is transferred from grassland to unused land. Meanwhile, related research also concluded that the grassland on the QTP is unstable, there is obvious degradation of alpine meadows, and natural grasslands have been degraded since the 1980s [74,75], and overgrazing also caused wide-scale grassland degradation on the QTP [76]. In northern Tibet, grassland degradation is very serious. Although the variability of precipitation has benefited the recovery and protection of the grasslands, temperature, and solar radiation variability exacerbated grassland degradation in Northern Tibet [77]. In addition, climatic changes also led to considerable degradation of alpine meadows and steppes in the western QTP. The surface layers of the grass soil became coarser and the water-holding capacity decreased [78].

Our results showed that over the past 20 years, the whole QTP showed an increasing trend in NDVI, which was consistent with the trend identified in other period (1982–2003 [79] and 2000–2019 [80]), indicating the vegetation activity on the whole QTP has been substantially enhanced. The NDVI trend of the QTP shows significant spatial differences [80,81]; we concluded that the NDVI-decreasing areas are mainly distributed in north Tibet, the alpine valleys in the southwestern QTP, and the Nu River in the southern parts of the plateau; while the NDVI-increasing areas are widely distributed throughout the QTP and mainly concentrated in the east and south, which was consistent with related studies [80]. Previous research also concluded the NDVI showed a downward trend in the south eastern area with rich hydrothermal conditions, while an increasing trend appeared in the northern plateau with poor hydrothermal conditions [30]. We concluded that NDVI in the northern and southwestern parts of the QTP is positively correlated with precipitation, while negative correlations are mainly distributed in the southeast of the QTP, including parts of Sichuan and Yunnan. The related studies also demonstrated that water availability was the main factor of vegetation growth in the northeastern and southwestern QTP [82]. The relatively dry environment in the northeastern and southwestern QTP limits the supply of water to vegetation growth [80], making the recovery and protection of the grasslands in these areas very sensitive to precipitation variability [77]. In contrast, summer precipitation was abundant in the southeast of the QTP. Increased precipitation was accompanied by an increase in clouds and, thus, a reduction in incoming solar radiation [83]. In addition, the increasing precipitation could contribute to soil erosion, which decreased soil organic matter content, and would restrict vegetation growth [84]. So, the NDVI in the southeast of the QTP showed a negative correlation with precipitation. Related studies have also found that the causes of the NDVI trend decline are different in

the QTP. In the central part of the plateau, a warming and drying climate caused a decrease in available water, which led to a downward trend in NDVI, while in the southeast areas, the downward trend in NDVI was mainly caused by a cooling and wetting climate [80].

Finally, due to the limitations of this study, we have not been able to carry out detailed quantitative research on the causes of periodic changes in precipitation and the main precipitation events in the region.

## 5. Conclusions

This study contributed to a better understanding of the characteristics of precipitation on the QTP during 1967–2016 based on the field observation data from meteorological monitoring stations, and its relationship with the LUCC was also discussed. The main conclusions of this study can be summarized as follows.

First, the precipitation of the whole QTP showed a significant increasing trend in the past 50 years with a rate of 11.5 mm/decade. On the seasonal scale, the increasing trend in summer was obvious; three periods (i.e., 22 years, 12 years, and 2 years) of oscillations in annual precipitation were observed; the four-season cycle scale was also concentrated at the three scales of 20–24, 10–12, and 2–4 years. Spatially, except for the 19.25% of areas in the southern and eastern parts of the QTP, which decreased significantly, most of the remaining areas showed an increasing trend. At the seasonal scale, such a trend was the most prominent in summer. The calculation results of the Hurst index showed that the indexes for the whole year and the four seasons were all greater than 0.5, indicating that the current changes had obvious persistence and that precipitation will show a consistently increasing trend.

Second, the most obvious land-use transfer between 1980 and 2018 occurred between grassland and unused land, and the mutual transfer area accounted for 22.79%; followed by the transfer between forests and grasslands, where the mutual transfer area accounted for 7.46%. From the perspective of spatial distribution, the transfer of grassland to unused land occurred in the western part of the QTP, while the reverse transfer was mainly distributed in the northwestern part of the QTP. Additionally, the mutual transfer between forest and grassland was mainly distributed in the southern and eastern parts of the QTP, while grassland shifts to woodland were also distributed in the western plateau. In addition, NDVI in the northern and southwestern parts of the QTP is positively correlated with precipitation, while negative correlations are mainly distributed in the southeast of the QTP, including parts of Sichuan and Yunnan.

Finally, the conclusions in this study would be valuable in the fields of regional hydrology, LUCC, and climate change.

**Author Contributions:** Conceptualization, B.Z.; methodology, B.Z.; software, B.Z.; validation, B.Z.; formal analysis, B.Z.; investigation, B.Z.; resources, B.Z.; data curation, B.Z.; writing—original draft preparation, B.Z.; writing—review and editing, B.Z.; visualization, B.Z.; supervision, W.Z.; project administration, W.Z.; funding acquisition, W.Z. All authors have read and agreed to the published version of the manuscript.

**Funding:** This research was funded by the National Nature Science Foundation of China (Grant No. 41977415)**.**

**Data Availability Statement:** The data presented in this study are openly available in [Data Center for Resources and Environmental Sciences, Chinese Academy of Sciences (RESDC), http://www.resdc.cn/DOI/doi.aspx?DOIid=54, accessed on 5 October 2020] at [DOI:10.12078/2018070201], reference number [34]; [Data Center for Resources and Environmental Sciences, Chinese Academy of Sciences (RESDC), http://www.resdc.cn/DOI/DOI.aspx?DOIid=49, accessed on 5 October 2020] at [DOI: 10.12078/ 2018060601], reference number [35].

**Acknowledgments:** We gratefully acknowledge financial support from the National Nature Science Foundation of China (Grant No. 41977415).

**Conflicts of Interest:** The authors declare no conflict of interest.

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
