# Peer review of "Spatial–Temporal Characteristics of Precipitation and Its Relationship with Land Use/Cover Change on the Qinghai-Tibet Plateau, China"

_land, doi:10.3390/land10030269_

Round 1
Reviewer 1 Report
The authors present an interesting investigation on spatial and temporal characteristics of precipitation and its association with land use/ land cover change (LUCC). The study is, however, a case study on the Qinghai-Tibet Plateau, China. Although the authors have conducted different popular methods (interpolation, trend detection, wavelet analysis, and correlation) and analyzed the long-term data, the mechanism's discussion is still weak. As I have previously mentioned during my earlier review, only presenting the obtained results is insufficient for a scientific paper.
The authors have used several methodologies in this paper. There is still a gap in why these methods were used and how they are interconnected. I still do not find why ANUSPLIN outperforms other approaches. Can you show or highlight the dependency of precipitation with the elevation field?
I find the revision is now improved a lot. I appreciate the in-depth study of spatial and temporal characteristics of precipitation and its relationship with LUCC. However, I feel there are some significant changes needed before it can be acceptable for publication. I compiled a list including recommendations and questions to the authors.
At first, the use of words like obvious should be done carefully. I was expecting some reasons behind the results on many occasions, but the authors explained them as "obviously". For me, it is not okay.
Abstract:
The LUCC change (1980 to 2018) shows decreasing grasslands (9.47%) and increasing bare lands (7.25%). If we relate these major changes, the decrease in NDVI should show these changes. In contrast, the yearly changes (1998 onwards) in precipitation and NDVI values show an increasing NDVI tendency. How do you relate these processes? A more in-depth discussion should be provided. And for a known fact and less detailed results, they can be removed from the abstract section.
I don't understand what do authors want to say from the following sentence. Also, the sentence is not grammatically correct. "will still immersed" ==> will still immerse or will still be immersed.
Line 19
It was predicted that QTP will still immersed in a pluvial stage.
Line 22 – 23
The shift between the two on the western plateau was obvious.
Why was the shift obvious? I do not find such a shift obvious.
Please choose the use of such words very carefully. It would be better to highlight the reasons behind the results rather than just saying the obvious.
Line 26: Qinghai-Tibet Plateauè ==> QTP
Introduction:
Line 35: Rainfall ==> Precipitation [ I suggest using the consistent word throughout either rainfall or precipitation, please check the entire manuscript.]
Line 50: Either the sentence needs a citation, or the authors should elaborate on why the uneven distribution of the rainfall stations attributes challenges to describe the spatial distribution.
Line 49 - 65
I do not think that the introduction section should have a more in-depth discussion on interpolation. Make it concise. If the paper's novelty is on interpolation, you need such a description in the introduction section. I suggest describing them in the method section and write concisely in the introduction section.
Results:
Line 240: Why the precipitation increased obviously?
Upon a close look at the temporal variation of precipitation and anomaly, one could find that the precipitation is higher than a long-term average since 2000. Assuming NDVI values and precipitation have a reasonable correlation, can we say that the NDVI values were low before 1998. Can you check to compute the NDVI values from 1980 to 1998 using older Landsat images to confirm this?
OR
Is it possible to analyze an intermediate land use cover map around 1998? Since the NDVI values are assessed only from 1998, the NDVI changes could help associate with Land use land cover.
Decreasing grasslands and increasing bare lands should associate with declining NDVI values. In contrast, the NDVI shows an increasing tendency, and it correlates with precipitation. In other locations, either there is a generally significant positive correlation or not significant positive correlation.
I could not link the results of wavelet analysis with other results. What can we expect from the periodicity analysis results in QTP? Apart from showing an increasing tendency of precipitation in the future, does the analysis show that the yearly variation might increase? Meaning a much drier year could follow the wetter year.
I do not see the following sentence is correct? I could see from figure 9 that negative correlations were observed across the south-eastern region of the study area.
Line 429-430
Fig. 9d shows that there was a large amount of negative correlation between the precipitation in the west and NDVI.
I do not understand the following sentence. Please elaborate on this.
Line 430 – 431
As precipitation increased, it affected the value of the NDVI to a certain extent, and Fig. 7 shows there was a clear grassland orientation.
Discussions:
The discussion section is not sufficient. There are significantly fewer discussions on the results that are presented.
Conclusion:
Overall the conclusion seems too concise. I suggest adding a paragraph and highlight the key results.
Reviewer 2 Report
Dear Authors,
I think that the paper is very valuable with a great deal of potential in the area of interpretation. I have made a number of comments related to editorial issues in a PDF file. My concerns focus on technical issues. Please check totals in Table 1 and in other tables. Please make uniform the range provided for axes – for example for graph 3. In addition, delete legends that repeat themselves. These are only suggestions. The conclusions in the paper are properly formulated.
Truly Yours,

Round 2
Reviewer 1 Report
Thank you for addressing most of the comments.
The revision is improved.
Author Response
Dear Reviewers:
On behalf of my co-authors, we appreciate reviewers very much for their positive and constructive comments and suggestions on our manuscript.